# The Role of Matrix Metalloproteinases in Thoracic Aortic Disease: Are They Indicators for the Pathogenesis of Dissections?

**DOI:** 10.3390/biomedicines12030619

**Published:** 2024-03-09

**Authors:** Marc Irqsusi, Lan Anh Dong, Fiona R. Rodepeter, Rabia Ramzan, Ildar Talipov, Tamer Ghazy, Madeline Günther, Sebastian Vogt, Ardawan J. Rastan

**Affiliations:** Heart Surgery University Hospital and Institute of Pathology, Philipps-University Marburg, 35037 Marburg, Germany; irqsusi@med.uni-marburg.de (M.I.); rodepete@med.uni-marburg.de (F.R.R.); ramzan@med.uni-marburg.de (R.R.); ildar.talipov@uk-gm.de (I.T.); tamer.ghazy@uk-gm.de (T.G.); madeline.guenther@gmx.de (M.G.); a.rastan@uk-gm.de (A.J.R.)

**Keywords:** metalloproteinases, aortic aneurysm, aortic dissection

## Abstract

The pathogenesis of aortic aneurysm and dissection continues to be under discussion. Extracellular matrix (ECM) remodeling processes in the aortic wall are hypothesized to be involved in the development of the disorders. Therefore, in a histological study, we investigated the expression of metalloproteases 1 and 9 (MMP1 and MMP9) and their inhibitors (TIMP 1 and TIMP 2) in cardiac surgery patients. In parallel, we studied the aortic roots by echocardiography. Clinical reports of 111 patients (30 women and 81 men) who suffered from aortic aneurysms and aortic dissection were evaluated and studied by transesophageal echocardiography. Seven patients who had coronary heart disease served as “healthy controls”. All patients underwent the necessary surgical procedure according to the diagnosed aortic disease in the period from 2007 to 2015. A tissue sample of the aortic biopsies was collected from each patient during surgery. Immunohistochemical staining was performed for MMP1 and MMP9 and TIMP1 and TIMP2 as well. Vascularization was monitored by a CD 31 antibody. In direct comparison, the expressions are not homogeneous. We found the smallest changes in the intima area at all. TIMP 1 and TIMP 2 distribution increases from the lumen of the vessel outward in the wall layers of the aorta. In the case of arteriosclerotic changes, intima had a capillarization, but not in the media. An opposite pattern was found in the dissected aortas. There are differences in the vascularization between the aneurysm and dissection and the different layers, respectively. A different remodeling process of the ECM in comparison to the vascular layers must be hypothesized. Reading the patterns of staining and with regard to the known inhibitory effect of MMP9 on ECM remodeling, but especially TIMP 2 on neoangiogenesis, disturbed nutrition, and dysfunctional vasa vasorum remodeling must be assumed as causes of dissection.

## 1. Introduction

Matrix metalloproteinases (MMPs) are a family of zinc-dependent extracellular matrix (ECM) remodeling endopeptidases. The tissue inhibitors of MMPs (TIMPs) inhibit the proteolytic activity of MMPs. Both of them modulate angiogenesis, cell proliferation, and apoptosis [1]. Recently, our group studied the MMP1 and 9 and TIMP 1 and 9 expressions, respectively, in the case of mitral valve insufficiency. We found a clear correlation between the MMP expression and the MI degree of severity [2,3]. The apparent close correlation between MMP expression and degeneration suggests a similar pathophysiological process in the formation of ascending aortic aneurysms (AAAs) and dissections. Therefore, the same set of protease/inhibitor antibodies was used immunohistochemically to study morphologic changes in the aortic wall during aneurysm formation or dissection. The investigations of metalloproteases and their inhibitors should provide information about the remodeling process of the ECM. It is assumed that this is not uniform and influences neoangiogenesis in different ways. It may result in disturbed tissue nutrition, which appears to be the basis for the development of aortic dissection. The focus was on three questions. On the one hand, it was important to show how the MMP/TIMP are distributed in the cross-section of the wall. On the other hand, when MMP/TIMP are distributed throughout the whole cross-section of the wall, the question remains whether MMP/TIMP patterns differ, either between the layers or between the aneurysm and dissection. At last, the question remains for insights into the pathophysiology of cardiovascular diseases. The available data are inconclusive. While MMP-1 and MMP-9 are reported to be elevated in thoracic chronic aneurysms, TIMP 1 and 2 are reported to be “similar to controls” or detected only to a low extent in aortic tissue [1,2].

## 2. Patients and Methods

### 2.1. Ethical Approval

This study was approved by the ethics committee of the Philipps-University of Marburg in September 2016 (Az.: 70/16).

### 2.2. Subjects

We examined the clinical reports of 111 patients (30 women and 81 men) who suffered from aortic aneurysms and aortic dissection (DeBakey I or STANFORD A), respectively. Herein, seven patients who had coronary heart disease served as “healthy controls” for histological examination. All patients were surgically operated on in the period from 2007 to 2015. The age of onset of patients ranged from 28 to 85 years, with a median age of 64 years. A tissue sample of the aortic biopsies was collected from each patient during surgery. The sample was fixed in a 10% formalin solution, embedded in paraffin, and archived at the Institute for Pathology at the Philipps-University of Marburg. Intraoperatively, appropriate fragments of the ascending aorta were taken and immediately histologically processed after fixation. Evaluation was performed by two colleagues with relevant experience in the field of vascular pathology.

### 2.3. Echocardiography

All patients underwent transoesophageal echocardiography immediately prior to sur gery. This involved measuring the aortic roots (Table 1) and examining the morphology of the aortic valves (Table 2). The competence of the mitral valves was demonstrated in all the examined groups.

### 2.4. Tissue Preparation for Staining

All the details of tissue preparation and staining were already described by Irqsusi et al. [3]. Sections were prepared using a sled microtome (SM2000 R; Leica, Wetzlar, Germany) and were mounted on glass slides (Thermo Fisher Scientific, Dreieich, Germany). The slides were placed in the incubator at 60 °C and were dewaxed for at least 60 min.

### 2.5. Hematoxylin and Eosin Staining

The tissue samples were stained with hematoxylin and eosin in a continuous linear stainer (Tissue Stainer COT 20, Medite Medical GmbH, Burgdorf, Germany). Xylene, alcohol (100%, 96%, and 70%), hemalum, eosin, alcohol (96% and 100%), and xylene were used to stain the slides in the given order. The tissue sections were subsequently assessed.

### 2.6. Elastica van Gieson Staining

For Elastica van Gieson staining, slides were incubated with resorcin fuchsin, water, HCl alcohol (10%), aqua dest., Fe hematoxylin (Weigert), aqua dest., van Gieson stain, ethanol (96%), ethanol (100%), and xylol in the given order.

### 2.7. Alcian Blue Staining

For staining, we used Alcian blue (1%), aqua dest., nuclear fast red solution, and aqua dest. and ethanol (70%, 96%, 100%).

### 2.8. Immunohistochemical Staining

The tissue slides were dewaxed by immersing in xylene, followed by ethanol (100%, 96%, and 70%) for 5 min each time. The slides were then immersed in the unmasking solution, placed in a steam cooker for 45 min, and rinsed with distilled water. The endogenous enzymes were blocked with a peroxidase-inhibiting solution and were rinsed with wash buffer. Primary antibodies were applied to the slides for 45 min (MMP-1: MAB 3307; Millipore, Burlington, MA, U.S.A.; MMP-9: NCL-MMP9-439; Leica Biosystems, Wetzlar, Germany; TIMP-1: M7293, Fa.Dako, Hamburg, Germany; TIMP-2: NCl-TIMP2-487; Leica Biosystems, Wetzlar, Germany). The slides were rinsed with wash buffer, followed by the application of secondary antibodies. The excess of secondary antibodies was removed with washing buffer, and the slides were immersed in Dako Real EnVision polymer conjugate for 20 min, followed by rewashing. The slides were then immersed in chromogen for 12 min and were rinsed with washing buffer and distilled water. They were counterstained with hematoxylin solution for 5 min. The tissue sections were then dehydrated with ethanol (70%, 96%, and 100%) to ensure complete dehydration. The slides were finally immersed in xylene. A few drops of entellan were poured on the tissue sections, which were covered with a coverslip. Afterward, the slides were examined under a light microscope (Olympus, Hamburg, Germany) with ×4, ×10, and ×40 magnification. The stained tissues were classified semi-quantitatively based on the percentage of stained cells. The following scale was used: 0 = negative (no visible color); 1 = positive in very few cells; 2 = positive in a moderate proportion of cells; and 3 = strongly positive for a larger proportion of cells. Positive control specimens of related antibody binding in histological examinations were performed according to Irqsusi et al. [3]. Detection of capillary vessels in the sections was performed by an anti-CD 31 antibody with the peroxidase staining technique (Monoclonal Mouse Anti-Human CD 31 for Endothelial Cells, Clone: JC70A, Dako Agilent (Santa Clara, CA, USA), M 0823, Diluted 1:50 with second AB against horse radish peroxidase (Dako Real EnVision HRP Rabbit/Mouse 5007)).

### 2.9. Imaging

The images were processed and recorded with a photomicroscope (Leica, Wetzlar, Germany) using Leica Im50 Software, version 4.0. Data from blinded specimens were evaluated by two independent, experienced pathologists. For morphometrical analysis, there was one section per marker/sample analyzed. Sections for different staining/immunohistochemistry were obtained as successive slides at a thickness of 2 µm. For an assessment, the whole slide was examined at a low magnification to find the area with highest marker expression. Slides were evaluated for comparison in a 100× magnification in these areas. For a score of 1 or higher, we expected to see any staining of at least medium intensity. For a score of 2, when evaluating at a field of 100× magnification, we expected a diffuse staining of medium intensity or a focal staining (<50%) of high intensity in the chosen field. A score of 3 corresponds to high-intensity staining of the marker in at least 50% field diameter.

### 2.10. Statistical Analysis

Statistical analysis was performed using IBM SPSS Statistics version 24 software, (IBM, Ehningen, Germany). *p* < 0.05 was considered significant for all statistical tests. For continuous variables, descriptive statistics (mean, median, minimum, and maximum) were used. Note, for categorical variables, frequencies (“*median rank”*) were evaluated to show the different appearance (“*frequency*”) and the extent (“*scores*”) of MMP and TIMP staining, respectively. Comparison of pairs from all groups was performed with the Krustal–Wallis test. Herein, we used the “*sample average rank*” of the groups. The groups were named “Aneurysms” (A), “Dissections” (D), and “Healthy controls” (H), corresponding to CIHD patients.

## 3. Results

### 3.1. Patients

Several patients were symptomatic at the time of diagnosis. Hyperlipidämia, diabetes mellitus type 2, and arterial hypertension were present among all the groups in proportions of 22.5%, 10.0%, and 74.8%, respectively. There were no differences between the groups (*p* = 0.813, *p* = 0.141 and *p* = 0.823). For further details, see Table 3. Comparing the groups, it is first noticeable that the patients with aortic dissection are somewhat younger than the patients who required surgery for aneurysm or CIHD. Both of the latter groups underwent predominantly elective surgery. In association with aneurysmal dilatation of the aorta, valvular insufficiency of the aortic valve was also prominent, requiring valve replacement. In the study group of dissections, a mortality of 21.2% had to be registered (Table 4). Moreover, in the aortic aneurysm group, an increased part of aortic valve replacements (AVRs) have to be mentioned (Table 4). When considering the concomitant diseases in our patient groups, it is noticeable that, apart from the comparison group with CIHD (*p* < 0.030), the proportion of coronary arteriosclerosis is also increased in the “aneurysm” group (Table 4). In contrast, no differences in the occurrence of arterial hypertension were found in all groups. Also of interest is that echocardiographic measurements of the aortic vessels show that the size of the sinutubular junction did not differ between the groups. The size of the aortic annulus, the ascending part of the aorta, as well as the maximum diameter were significantly smaller in the patients with dissection compared with the aneurysm group (Table 2).

### 3.2. Histological Examinations

The first important point to note is that there is no uniform distribution among the target proteins detected, either between the groups or between the wall layers of the aorta in the groups. In the small comparison group of patients with CIHD (punch cylinder of the ascending aorta before implantation of bypasses), there are weak signs of MMP1 enzyme expression at the intima (grade 1, 28.6%) and throughout the media (grade 1, 100%). MMP 9 was expressed at the intima in 14.3% only. Regarding corresponding data according to the original counts (see Table 5 and Table 6) for proteinase inhibitors (TIMP 1 and TIMP 2), we found no staining across all tissue layers in the comparison group.

Opposite effects were found in the aneurysmatic vessel group and in the group with dissected vessels, as well. Interestingly, consideration of matrix metalloprotease distribution and their inhibitors, both by type and by localization in the vessel wall, does not give a uniform impression. While on the one hand, the effect of these enzymes and their counterparts, the inhibitors, is a dynamic process, on the other hand, the histological examination represents only a temporal section of this process; we speak in the following of “expressions” in order to emphasize the “actual state”. The change in the quantities in the dynamics of the pathophysiological process is highly probable but cannot be investigated within the scope of this work. Histological sections of the corresponding aortic fragments were taken intraoperatively and stained. In the evaluation, a score assignment of the staining intensities was performed, and their frequencies of occurrence in the groups were statistically processed. According to the lack of normal distribution of the parameters, the median rank values in the aneurysm (A) and dissection (D) groups were first compared. The MMP 1 expression in the intima and media of the aortas show a rather balanced distribution of the frequencies (standards for immunostains by the corresponding ABs, Figure 1; examples for histological grading and subsequent scores, Figure 2). In the adventitia, we found a shift to higher scores in cases of dissection. For MMP 9, we found a balanced distribution in the intima, with a tendency toward higher scores in the media and adventitia according to the median rank distribution (Figure 3A–D). For the inhibitors, TIMP 1 shows a balanced distribution in the intima and a tendency toward higher scores and frequencies in the media and adventitia. In contrast, TIMP 2 shows a balanced pattern in all vessel sections. It is indeed surprising that a closer look at the HE staining of the specimens does not reveal a differentiation between media degeneration and progressive arteriosclerosis (see Table 7 and Table 8 and Figure 4 and Figure 5).

For further analysis, the sample average ranks of the groups (aneurysm (A), dissection (D), and the comparison group (H)) were compared in a triangle scheme, and the significances were calculated exactly. Strikingly, there is a large difference between dissection and aneurysm in the MMP 1 expression of the adventitia (*p* = 0.001). MMP 9 expression in the media and adventitia also differed between these two groups. However, differences in MMP 1 and MMP 9 in the intima could not be shown (Figure 6A). We found a clear difference in TIMP 1 expression in comparison to the control group, both in the intima of aneurysm patients and in the media of patients with a dissection (Figure 6B). TIMP 1 was found to be significantly increased in the adventitia in the case of dissection compared with aneurysmal tissue, but differences from the control group could not be detected because of the variance of the results. On the other hand, TIMP 2 showed clear differences in the intima and adventitia, but TIMP2 was found to increase in the intima in the case of aneurysm and TIMP2 in the adventitia in the case of dissection. Detection of TIMP2 in the media did not reveal any differences.

Some notable changes in the histological evidence of metalloproteases and antagonists were compiled in a synopsis. The grading-specific data (see Table 5 and Table 6) were used. Figure 7 shows important changes in MMP1 and MMP 9 corresponding to the layers of the aorta. It can be seen that MMP 1 and MMP 9 expressions increase significantly in the adventitia in the case of dissection (higher grading). MMP 9 is also increased in the media. There are no significant changes in metalloproteases in the intima. With regard to the protease inhibitors, the changes for the grade 2 values of TIMP 1 and grade 1 values of TIMP 2 are shown in Figure 8. The rather mild expression of the latter parameter in the intima and media encountered in the aneurysmal changes is found in the adventitia in the case of dissection. A stronger expression of TIMP 1 (grade 2) is found in aortic dissections. The expressions of TIMP 1 increase significantly from the vessel lumen to the adventitia. In the case of an aneurysm, this kind of expression has an opposite direction.

TIMP1 is found to be increased in the vessel wall of aortic aneurysms, but in the intima, it is stimulated by TNFα. In our study, overall, the highest values of TIMP1 expression were detected not in the case of dissection but in the adventitia. A difference was seen in the media and in the adventitia but not in the intima. Interestingly, there are no differences in the description of media-generative or arteriosclerotic alterations in the histological examination (Figure 4 and Figure 5, Table 7 and Table 8), but staining the vessels opens a new view of the sections. Although in the case of aneurysms, vascularization is found dominantly in the intima, the opposite effect appears in medial degeneration and dissection, whereas many vessels can be detected in media and adventitia (Figure 9).

## 4. Discussion

Three questions remain to be answered. If rupture of the intima from the inside is to be assumed as a pathological substrate for the development of aortic dissections, it could also be assumed that metabolism shows a certain uniformity of changes from the inside to the outside with respect to the vascular layers. The present concept holds that an increased intraluminal pressure (arterial hypertension) initially causes a dilatation of the aortic tube, and finally, a rupture of the intima is provoked by mechanical shear forces, leading to dissection. This assumption is contradicted by two facts of this study: on the one hand, there are clear differences in the distribution of proteases and their inhibitors in the layers (see Figure 3A–D and Figure 6A,B and Table 5 and Table 6), and on the other hand, the echocardiographic studies clearly show that the native diameters of aortic dissections are smaller than those of aneurysms (Table 3).

Hence, it cannot be intraluminal pressure alone that leads to the tearing of a damaged intima and allows intimomedial blood flow. There must be indwell changes in the wall, i.e., material-dependent causes for aortic dissection. Extracellular matrix remodeling occurs in both blood vessels and the heart in response to increases in mechanical forces, such as raised blood pressure and flow [5].

There is an interplay between cells and the extracellular matrix. Cardiovascular aging is a physiological process affecting all components of the heart and the vessels. Increasing evidence points to the pivotal role of the extracellular matrix (ECM) [3,6]. Coming back to the first question. When proteases and their inhibitors are differentially distributed throughout the vessel wall, a different status of metabolism can be suggested.

### 4.1. MMP and TIMP as Mediators for ECM Modelling

Blood vessels are exposed to several forms of mechanical force—shear stress, pressure, and tensile stress. The latter has several components, including circumferential stress caused by expansion or dilation of the vessel wall and internal stresses generated by the cells themselves in response to the external forces. Out of the collagenases, we have selected MMP 1. Collagenases (MMP-1, MMP-8, MMP-13, MMP-18) are hydrolytic enzymes acting on collagen types I, II, and III found between cells. The product of catabolism of these substances is obtained from denatured collagen or gelatin. They can also act on other molecules of the extracellular matrix. From the gelatinases (MMP-2, MMP-9), we studied MMP-9. They help in the catabolism of type I, II, and III collagens. They also act on denatured collagen or gelatin obtained after the action of collagenases [6]. As counterparts of the proteases, we examined TIMP1 and TIMP2. MMP 1 expression is associated with increased rates of aneurysmal rupture and reduced survival and aortic dissection. The expression is stimulated by TNFα and IL1 [7,8]. TNF-α signaling triggers downstream epigenetic modifications that result in lasting enhancement of pro-inflammatory responses in cells. For example, it is a potent chemoattractant for neutrophils and promotes the expression of adhesion molecules on endothelial cells, helping neutrophils to migrate. According to the results of our study, there is no difference in MMP1 expression between the intima and the media, either in the case of an aneurysm or dissection or in comparison to the control group. The highest levels of MMP1 expression were found in the adventitia of the patient group with a dissection. Thus, it must be assumed that only perivascular inflammatory processes favor the formation of an aortic dissection in contrast to aneurysms. This result is supported by the observation of LaMaire et al., according to their experimental work. Compared with aortic tissues from mice that received ciprofloxacin, they showed decreased expression of lysyl oxidase, an enzyme that is critical in the assembly and stabilization of elastic fibers and collagen. He concluded that ciprofloxacin increases susceptibility to aortic dissection and rupture in a mouse model of moderate, sporadic aortic dissections [9,10]. Ciprofloxacin is involved in the breakdown of ECM in inflammatory processes, dissolving interstitial collages by induction of MMP1 [11]. MMP9 appears associated with the development of aortic aneurysms, because its suppression prevents the development of aortic aneurysms. Interestingly, doxycycline, contrary to ciprofloxacin, suppresses the growth of aortic aneurysms in animal models through its inhibition of MMP 9 by reducing aortic inflammation in humans [11,12,13]. MMP-9 or gelatinase B, together with MMP 2, belongs to the key effectors of ECM remodeling and is greatly upregulated in wound repair. Based on these few facts, it can be stated with regard to the first question that active remodeling processes of the ECM primarily take place in the wall of dissecting vessels as a direct response to a mechanical impact.

When asking the second question, how does the expression of the investigated parameters differ in the wall layers, the function of the proteases and their inhibitors must be considered. MMP 9 appears to be a regulatory factor in neutrophil migration [14] and may play an important role in angiogenesis and neovascularization. It is a key regulator of both growth plate formation and growth plate angiogenesis. Increased expression of MMP9 (e.g., in the heart and cardiovascular system) means an expression of a disease state and is not part of the normal expression pattern of various tissues [1,14]. MMP 2 and MMP9 work together in ECM remodeling, vascular smooth cell apoptosis, balancing ROS, and triggering inflammation [15]. The highest values for MMP9 expression were found in the case of dissection. No differences were found in the intima, but the media and, again, the adventitia had an extreme increase in the marker. This could mean a defense for the vascular wall to protect from pressure loading and an indicator for maximum TGF-β activation. Therefore, the involvement of MMP 9, along with MMP 2, in the formation of aneurysms may be derived [16]. Infiltrating leukocytes, cardiomyocytes, fibroblasts, and endothelial cells secrete MMP-9 during all phases of tissue repair. MMP-9 both exacerbates the inflammatory response and aids in inflammation resolution by stimulating the pro-inflammatory to reparative cell transition. In addition, MMP-9 has a dual effect on neovascularization [17]. In the case of dissection, we found an increase in the expression of TIMP1 and 2 from “inside to outside”. The highest values are found in the adventitia. The glycoproteins are natural inhibitors of MMP1 and MMP 9 that are able to promote cell proliferation in a wide range of cell types and may also have an anti-apoptotic function [18]. They are stimulated by TNFα. In our study, overall, the highest values of TIMP1 expression were detected in the case of dissection. This difference was seen in the media and in the adventitia but not in the intima. TIMP2 is a peptidase involved in the degradation of the extracellular matrix. In addition to an inhibitory role against metalloproteinases, the encoded protein has a unique role among TIMP family members, having the ability to suppress the proliferation of endothelial cells, directly. This protein may be critical to the maintenance of tissue homeostasis by suppressing the proliferation of quiescent tissues in response to angiogenic factors and by inhibiting protease activity in tissues undergoing remodeling of the extracellular matrix [19,20,21]. Interestingly, the highest TIMP2 expression was measured in the intima in the case of aneurysms. TIMP2 was less expressed in the case of dissection. Conversely, we found higher expression in the adventitia in the case of dissection and, to a lesser extent, in the case of aneurysm. No differences were found in the media. In the investigated aortic tissue, a clear proteolytic transformation can be assumed, which favors progressive destruction of the extracellular matrix, but degradative and constructive (reparative) processes can also be assumed, which focus on angiogenesis and neoangiogenesis of the supplying blood vessels (vasa vasorum) [2,22].

### 4.2. It Is All about the Vessels of the Vessels

The third question addressed the differences between the proteases and inhibitors in their expressions regarding conclusions about assumed pathomechanisms for vascular dissections. Angiogenesis can be directly or indirectly mediated by MMPs through the modulation of the balance between pro- and anti-angiogenic factors [23]. The role of vasa vasorum (VV) in atherosclerosis is controversial; experimental and clinical evidence strongly suggests the potential of VV in vascular proliferative disorders [24]. Since vasa vasorum are end arteries, they easily develop hypoxia and/or ischemia in the cells of the intima or media of the arterial wall. They are also known to be the most common sites of atherosclerosis [25]. The recruitment and accumulation of inflammatory cells and the subsequent release of several cytokines, especially from resident macrophages, stimulate the expansion of existing VV and the formation of new highly permeable microvessels. So, angiogenesis of VV appears responsible for the initiation and progression of atherosclerosis [26]. Atherosclerosis is initiated by endothelium activation and, followed by a cascade of events (accumulation of lipids, fibrous elements, and calcification), triggers the vessel narrowing and activation of inflammatory pathways [27]. The vasa vasorum forms a network of microvasculature that originates primarily in the adventitial layer of large arteries. These vessels supply oxygen and nutrients to the outer layers of the arterial wall. The expansion of the vasa vasorum to the second order is associated with neovascularization related to the progression of atherosclerosis [28]. We found increased TIMP1 and TIMP 2 expression in the adventitia of dissected aortic tissue markers, which indicates inhibition of neoangiogenesis [22].

### 4.3. Modification of the Extracellular Matrix by HIF

When collagen proteins (MMP-1, -8, and -13) are associated with angiogenesis, and their loss results in irreversible rupture of the matrix and TIMP-1 and TIMP-2 regulate type IV collagen and their participation in cell endothelial migration in the interstitial stromal spaces, then it is highly probable that they are playing a key role in angiogenesis regulation by inhibiting neovascularization [23]. Absolutely consistent, Billaud et al. reported medial hypoxia and vasa vasorum remodeling in the aneurysmal ascending aorta. They found that aneurysmal tissue is characterized by a lower density of small-size vasa vasorum [29]. They point out that these data highlight differences in vasa vasorum remodeling and associated medial chronic hypoxia markers between aneurysms of different etiology. These aberrations could contribute to malnourishment of the aortic media and could conceivably participate in the pathogenesis of the thoracic aortic aneurysm. Supporting this statement, Son et al. histologically demonstrated a complete regional absence of vasa vasorum in an aneurysm of the ascending aorta [30]. Even in our histological study, media and adventitia regions appeared empty regarding vasculogenesis. TNF-α, IL-8, and other factors with a known pro-angiogenic capacity stimulate the production of MMP-9 in endothelial cells and regulate the angiogenesis process [23]. In our study, we found a higher MMP1 and MMP 9 expression in the adventitia but also a higher MMP9 expression in the media fitting to the already mentioned remodeling processes in the aortic wall. Increasing parts of TIMP1 and TIMP 2 expression from “inside to outside” (intima/media/adventitia) indicate affected neoangiogenesis of VV in the case of dissection and prove that this finding is a possible characteristic histological marker. The increased evidence of MMP9 expression in the media (induced by several angiogenic factors) and TIMP 2 expression within the adventitia (suppressed neoangiogenesis) indicates an important biomechanical factor leading to dissection. The diameters of the dissected aortas were smaller in the study group, and thus, according to Laplace’s law, the surface tension of the aortic wall should have been lower than in the group of patients who had an aneurysm than in dissected vessels.

### 4.4. Aspect of Biomechanics

By using the finite element approach for the identification of the anisotropic hyperelastic properties of normal and diseased aortic walls, we found a much higher cyclic longitudinal strain in the ascending aorta vs. abdominal aorta than in the circumferential strain [31,32]. Moreover, absolute values of the cyclic rotation amplitudes of the ascending aorta are also much higher than those of the abdominal aorta. The ascending aorta undergoes a complex deformation with alternating clockwise and counterclockwise twists. Longitudinal strain and its phase shift to circumferential strain contribute to the proximal aorta’s Windkessel function. Complex cyclic deformations are known to be highly fatiguing. This may account for the increased degradation of components of the aortic wall and, therefore, promote aortic dissection or aneurysm formation. Moreover, a strong negative correlation between flow-induced vascular stress and media thickness was observed. Li et al. have shown that vessel structural stress mediates aortic media degeneration [33]. These statements correlate with the generally accepted view that rupture or hemorrhage of the vasa vasorum indicates the initial moment for dissection.

### 4.5. Conclusions

MMPs and their inhibitors are involved in cell repair and the remodeling of tissues of the ECM. They are induced by mechanical load. Data evaluation of our immunohistological study allows us to make an assumption about the cause of aortic dissections, at least in the ascending part of the aorta, due to the increasing distress between longitudinal and circumferential strain or deformation [31,32], respectively, resulting from an increase in diameter of the ascending aorta, which in turn is caused by hypoxia or HIF-mediated remodeling of the extracellular matrix. This results in mechanical stress. Mechanical support by cyclic amplitudes leading to aspiration and squeezing of the medial blood in and out of the aortic wall for the perfusion of the vascular media is lost. Consequently, under hypoxic conditions, new vessels are induced by the interaction of HIF1α and VEGF and inflammation. But they are too small and may be malformed [29,34]. The accumulated knowledge about the importance of metalloproteases and their inhibitors, as well as their overall involvement in metabolism, seems to justify the assumption of the intrinsic processes of the aortic wall for the development of dissection [35]. This “outside-in mechanism” has already been formulated, wherein aortic dissection has been characterized as a disease of the nourishing vessels [36].

### 4.6. Limitations

The present study is a clinically randomized study, which means there is a retrospective inclusion of patients in a certain time period. While the majority of surgical interventions for aortic aneurysm repair or aortocoronary bypass grafting were performed as elective procedures, the majority of aortic dissection surgeries were emergency procedures (Table 1). Although the gender composition in the individual groups did not differ, the results of the study should be confirmed in a study with uniform gender proportions. In fact, the extent of arteriosclerotic lesions along the aorta can change. Likewise, cystic media degeneration in the aorta is not ubiquitous. This is certainly a limitation of histologic studies. The tissue samples were taken during the surgical procedure and are most likely representative based on clinical experience. Last but not least, other MMPs should also be included in the study. The question of vascularization or neovascularization (see [30]) remains an important subject of investigation to support the observations of our study.

## Figures and Tables

**Figure 1 biomedicines-12-00619-f001:**
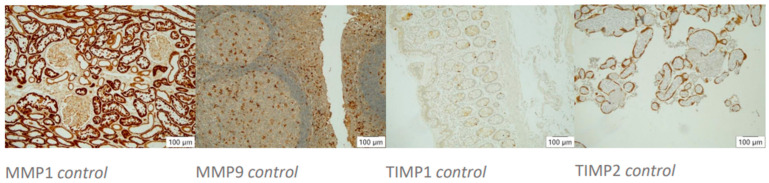
Histocontrols of AB’s action. Positive controls (10× magnification, left to right): MMP1 stains tubular epithelia in kidney tissue, MMP9 stains diffusely distributed macrophages in tonsil tissue, TIMP1 stains single inflammatory cells in colonic mucosa, TIMP2 stains trophoblast cells in placental tissue.

**Figure 2 biomedicines-12-00619-f002:**
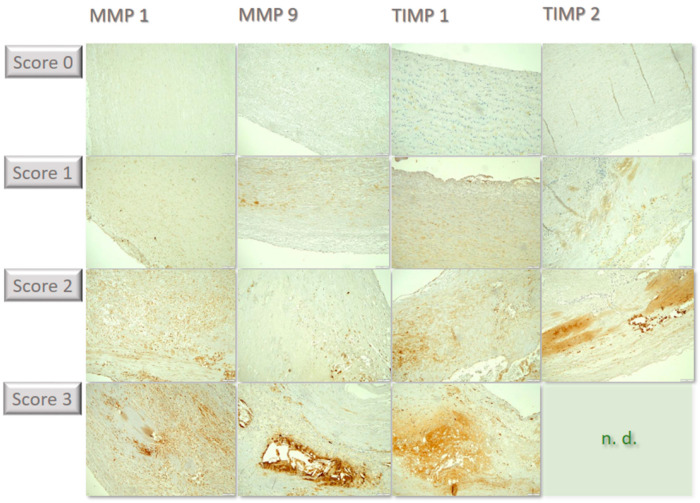
Compilation of individual examples of the differential immunochemical staining in the tissue sections. MMP1 was lowly expressed, corresponding to score of 0. In some specimens, few cells showed staining, corresponding to score of 1. In some cases, staining becomes positive in a moderate proportion of cells, corresponding to score of 2; in the example, cells are predominantly found within atherosclerotic plaques in the lower portion of the image. For MMP 9, a few samples were stained strongly with many cells, corresponding to score of 3, such as in the example above within an atherosclerotic plaque grouped around flushed-out cholesterol needles. In some cases, staining of TIMP 1 was seen in few individual cells, corresponding to score of 1, such as in the above example in small superficial lipid spots. Other samples showed expression in a moderate proportion of cells, corresponding to score of 2, including in the cell-rich areas of inflammatory activity in the media. Some cases showed labeling for TIMP2 in a moderate number of cells, corresponding to score of 2, such as in the above example within an intimal plaque. Strong positivity in a large proportion of cells (score of 3) could not be detected in any sample (n.d.). All immunohistological images were standardized and taken at 100× magnification.

**Figure 3 biomedicines-12-00619-f003:**
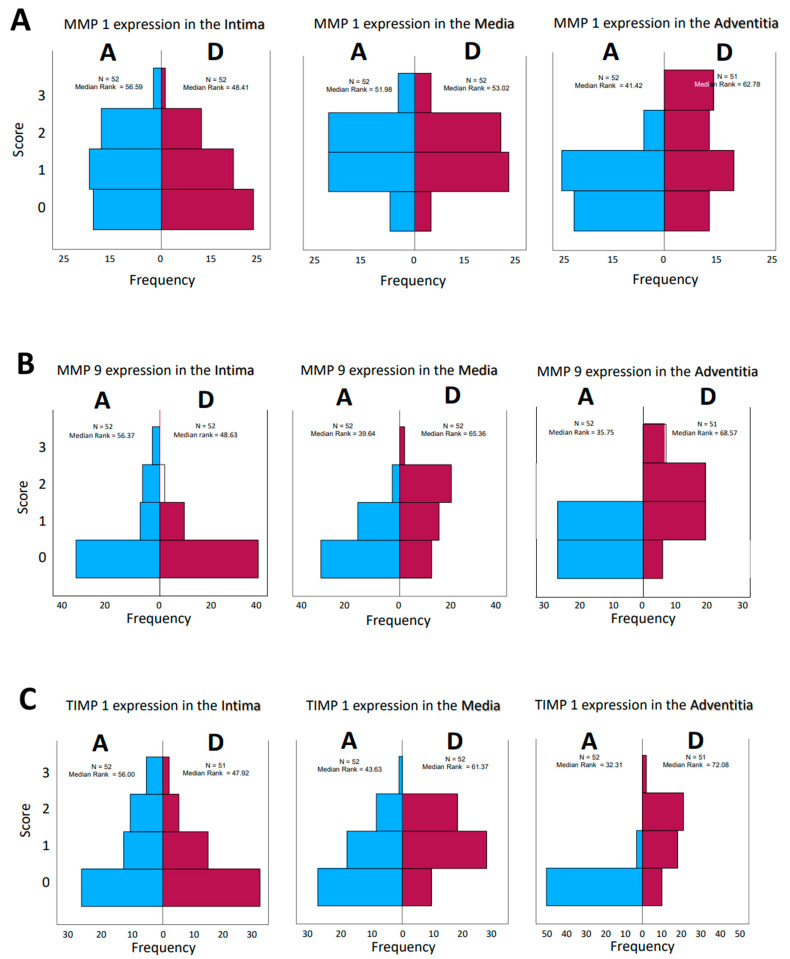
(**A**–**D**): Frequencies of encountered scores in histological evaluation of tissue sections from patients who had aortic aneurysm (A) or aortic dissection (D). The values for the median rank were entered. The results for MMP 1 and MMP 9 and TIMP1 and TIMP 2 were presented separately for the tissue layers’ intima, media, and adventitia, respectively.

**Figure 4 biomedicines-12-00619-f004:**
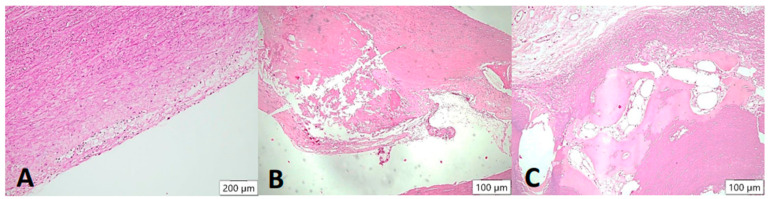
Arteriosclerotic lesions in the aortic wall (HE stain). In the samples examined, the degree of atherosclerosis varied markedly from mild to severe cases. In the initial plaque, a few foam cells can be demarcated in the intimal layer (**A**), which may eventually appear macroscopically together with smooth muscle proliferates as so-called “fatty streaks”. In the advanced atheromatous bed (**B**), smooth muscle proliferates are more prominent in the arterial wall. In addition, abundant cholesterol-containing material can be seen. In isolated cases, heterotopic ossification with formation of bone bellows and intervening medullary spaces was seen in the plaque (**C**).

**Figure 5 biomedicines-12-00619-f005:**
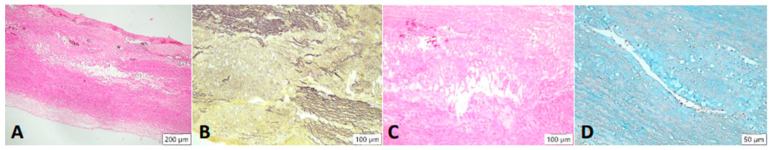
Cystic degeneration of the media. Lesions are found mainly within the media of the vessel: visually empty gaps in the tissue are seen in the HE stain, where the fibers break apart ((**A**), 4× magnification). In the Elastica van Gieson stain, fragmentation of elastic fibers is highlighted ((**B**), 4× magnification). These changes are shown at 20× magnification. An incorporation of basophilic ground substance is seen in the gaps of the vessel wall ((**C**), HE stain), which contains mucopolysaccharides. Band can be detected in the Alcian blue stain ((**D**)).

**Figure 6 biomedicines-12-00619-f006:**
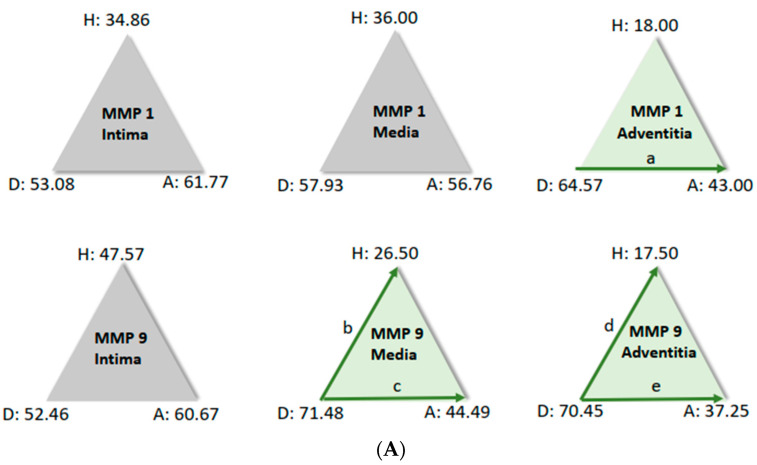
(**A**) Comparison of pairs from all groups’ MMP1 and MMP9 (Kruskal–Wallis-Test); each node shows the sample average rank of the group. Test was considered “n.s.” when *p* ≥ 0.05; significance is shown with green arrows. a = 0.0001; b = 0.001; c = 0.0001; d = 0.031; e = 0.0001. H = healthy; D = dissection; A = aneurysm. (**B**) Comparison of pairs from all groups’ TIMP1 and TIMP9 (Kruskal–Wallis-Test); each node shows the sample average rank of the group. Test was considered “n.s.” when *p* ≥ 0.05; significance is shown with green arrows. a = 0.029; b = 0.0001; c = 0.004; d = 0.0001; e = 0.0001; f = 0.030. H = healthy; D = dissection; A = aneurysm.

**Figure 7 biomedicines-12-00619-f007:**
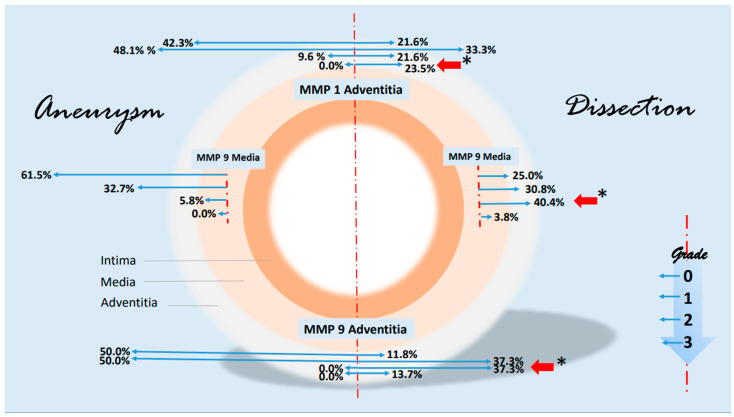
Synopsis showing partial results of MMP1 and MMP9 expressions in the histological examinations. Data are given in % grading proportions in relation to the respective study groups. In the case of aneurysm, we find high rates of low-grade staining in MMP1 and MMP 9 expression in the adventitia. In contrast, the proportions shift in the assessment of dissection. Both low-grade and high-grade staining are significantly increased in adventitia (red arrows, * *p* < 0.001), highlighting the important role of MMP 1 and MMP 9 in the remodeling pathomechanism, possibly on an inflammatory basis. This kind of shift in expression is particularly evident for MMP 9 in the media. The schematic picture of an aortic cross-section is divided into two parts. On the left side, some findings of MMP 1 and MMP 9 staining in aneurysms are shown. On the right side, corresponding findings of aortic dissections were presented. Starting from the interrupted red line, the extent (or size) of the staining is shown in blue arrows to the left and right according to the pathology. The grading, which corresponds to the intensity of the staining, is plotted progressively from top to bottom (see small diagram below right). The results are shown in relation to the adventitia and media. For instance, we find a difference in MMP 9 expression grade 2 in the media of 40.4% (red arrow) in a dissection versus 5.8% in an aneurysmatic vessel or 37.3% grade 2 expression of MMP 9 in the adventitia of a dissection versus 0.0% in aneurysmata or a 23.5% grade 3 increased expression of MMP1 in the adventitia (* *p* < 0.001), respectively. The basis for this presentation is direct immunohistologic evaluations of 2 independent pathologists in the detection of MMP 1 and 9 (see Table 5).

**Figure 8 biomedicines-12-00619-f008:**
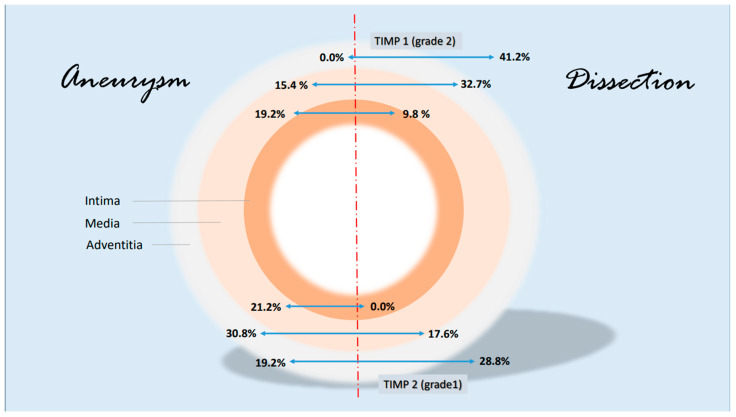
Synopsis showing partial results of TIMP 1 and TIMP 2 expressions in the histological examinations; presentation chosen resembles Figure 7. Data are given in % grading proportions in relation to the respective study groups. The protease inhibitors are reduced at maximum in the intima in case of a dissection. This allows two hypotheses. On the one hand, this may indicate an increased remodeling process in response to increased intraluminal pressure in the aorta and damage in the media (compare increased MMP 9 values, synopsis Figure 7) or increased neocapillarization, which, however, decreases from the “inside to the outside” of the aortic lumen. Grade 3 staining was mild and showed no difference between groups. Increasing parts of TIMP1 and TIMP 2 expression from “inside to outside” (intima/media/adventitia) indicate affected neoangiogenesis of VV in case of dissection and uncover this finding as possible characteristic histological marker.

**Figure 9 biomedicines-12-00619-f009:**
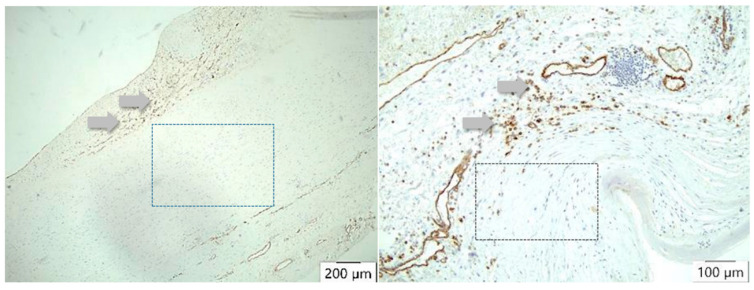
CD 31 antibody/peroxidase staining technique of the vessels. In atherosclerosis, neovascularization (arrows) occurs mainly in the intima (**left**), whereas in case of dissection (**right**), many vessels can be detected either in the media or adventitia (arrows) but not in the intima. There is no staining of vessels in the arteriosclerotic media (left square) or in the dissected aortic media (right square).

**Table 1 biomedicines-12-00619-t001:** Echocardiographic measurements of the aortic roots (data in mm). Bold numbers represent significance. The parameter in concern has been marked with an asterisk.

Diameter	Aneurysm	Dissection	Total	*p* Value
Anulus, Median Range	34.6 * 22.5–50.2	29.1 23.2–39.9	30.8 22.5–50.2	**0.006**
S.-tubular Junct., Median/Range	47.4 32.3–61.2	41.8 29.4–59.0	45.1 38.1–50.6	0.272
Ascending Part Median/Range	50.3 * 39.6–67.0	44.4 33.1–59.0	48.3 33.1–67.0	**0.003**
Max. Diameter Median /Range	51.5 * 39.6–67.0	46.7 33.1–59.0	49.6 33.1–67.0	**0.003**

**Table 2 biomedicines-12-00619-t002:** Echocardiographic findings of the aorta in the study groups *. The parameter in concern has been marked with an asterisk.

Aortic Valve Stenosis	Aneurysm	Dissection	*p* Value
No Stenosis	41 (80.4%)	47 (94.0%) *	**0.038**
Grade I	4 (7.8%)	2 (4.0%)	
Grade II	0 (0.0%)	0 (0.0%)	
Grade III	6 (11.8%)	1 (2.0%)	
**Aortic Valve** **Insufficiency**			
No Insufficiency	6 (11.8%)	17 (34.7%)	**0.023**
Grade I	11 (21.6%)	10 (20.4%)	
Grade I–II	6 (11.8%)	1 (2.0%)	
Grade II	8 (15.7%)	7 (14.0%)	
Grade II–III	3 (5.9%)	6 (12.2%)	
Grade III	14 (27.5%)	5 (10.2%)	
Grade III–IV	3 (5.9%)	2 (4.1%)	
Grade IV	0 (0.0%)	1 (2.0%)	
**Aortic Valve** **Morphology**			
Bicuspid Valve	3 (5.9%)	4 (8.0%)	
Biscupid-Like Valve	13 (25.5%)	1 (2.0%)	
Tricuspid Valve	35 (68.6%)	45 (90.0%) *	**0.003**

* in the aortic aneurysmatic group, 1 (1.9%), and in case of dissection, 2 (3.8%) of the patients had already an preexisting aortic valve replacement. In the “healthy” group, no insufficiency or stenosis of the tricuspid valves were found.

**Table 3 biomedicines-12-00619-t003:** Concomitant diseases in %. Bold numbers represent significance. The parameter in concern has been marked with an asterisk.

Concomitant Diseases	Aneurysm	Dissection	Control	Total	*p* Value
Hyperlipidemia	23.1	21.2	28.6	22.5	0.813
Arterial Hypertension	73.1	75.0	85.7	74.8	0.823
CIHD	38.5	19.2	85.7 *	32.4	**0.030**
Diabetes mellitus II	11.5	3.8	28.6	9.0	0.141

**Table 4 biomedicines-12-00619-t004:** Patients’ data. Bold numbers represent significance. The parameter in concern has been marked with an asterisk.

	Aneurysm (*n* = 52)	Dissection (*n* = 52)	Control (*n* = 7)	Total (*n* = 111)	*p* Value
**Gender**					
male	40 (76.9%)	36 (69.2%)	5 (71.4%)	81 (73.0%)	0.377
female	12 (23.1%)	16 (30.8%)	2 (28.6%)	30 (27.0%)	0.377
**Age/Median**	65.0 years	60.0 years	78.0 years	64.0 years	**0.011**
Range	40.0–84.0 years	28.0–82.0 years	51.0–85.0 years	28.0–85.0 years	
**EF**					
normal	40 (76.9%)	36 (70.6%)	4 (80.0%)	80 (74.1%)	0.337
reduced	5 (9.6%)	7 (13.8%)	0 (0.0%)	12 (11.1%)	0.539
Elective surgery	51 (98.1%) *	1 (1.9%)	5 (71.4%)	57 (51.4%)	**<0.001**
emergency	1 (1.9%)	51 (98.1%) *	2 (28.6%)	54 (48.6%)	**<0.001**
**Surgical procedure**					
valve- type conduit	10 (19.2%)	6 (11.5%)	0 (0.0%)	16 (14.4%)	0.277
supracoronary replacement	40 (76.9%)	31 (59.6%)	0 (0.0%)	71 (64.0%)	0.058
supracoronary ascending aorta replacement with arch replacement	2 (3.8%)	15 (28.8%) *	0 (0.0%)	17 (15.3%)	**<0.001**
**additional surgical procedures**					
**AVR**	21 (40.4%) *	3 (5.8%)	0 (0.0%)	24 (21.6%)	**<0.001**
**AVRepair**	13 (25.0%)	12 (23.1%)	0 (0.0%)	25 (22.5%)	0.697
**CABG**	10 (19.2%)	7 (13.4%)	5 (71.4%)	22 (19.8%)	0.239
**Complications**					
Arrythmia	9 (17.3%)	4 (7.7%)	0 (0.0%)	13 (11.7%)	0.138
Bleeding	7 (13.5%)	10 (19.2%)	0 (0.0%)	17 (15.3%	0.426
Infection	2 (3.8%)	5 (9.6%)	0 (0.0%)	7 (6.3%)	0.240
Ischemia	3 (5.8%)	5 (9.6%)	0 (0.0%)	8 (7.2%)	0.462
Organ failure	1 (1.9%)	1 (1.9%)	0 (0.0%)	2 (1.8%)	1.000
**Mortality**	2 (3.8%)	11 (21.2%) *	0 (0.0%)	15 (13.5%)	**0.008**

**Table 5 biomedicines-12-00619-t005:** Exact numbers of MMP-1 and -9 grading data by two independent pathologists for the three groups (mean values from two independent but concordant evaluations).

	A (*n* = 52)	D (*n* = 52)	H (*n* = 7)	Total (*n* = 111)
**MMP1. Intima**				
0	17 (32.7%)	23 (44.2%)	5 (71.4%)	45 (40.5%)
1	18 (34.6%)	18 (34.6%)	2 (28.6%)	38 (34.2%)
2	15 (28.8%)	10 (19.2%)	0 (0.0%)	25 (22.5%)
3	2 (3.8%)	1 (1.9%)	0 (0.0%)	3 (2.7%)
**MMP1. Media**				
0	6 (11.5%)	4 (7.7%)	0 (0.0%)	10 (9.0%)
1	21 (40.4%)	23 (44.2%)	7 (100.0%)	51 (45.9%)
2	21 (40.4%)	21 (40.4%)	0 (0.0%)	42 (37.8%)
3	4 (7.7%)	4 (7.7%)	0 (0.0%)	8 (7.2%)
**MMP1. Adventitia**				
N-Miss	0	1	5	6
0	22 (42.3%)	11 (21.6%)	2 (100.0%)	35 (33.3%)
1	25 (48.1%)	17 (33.3%)	0 (0.0%)	42 (40.0%)
2	5 (9.6%)	11 (21.6%)	0 (0.0%)	16 (15.2%)
3	0 (0.0%)	12 (23.5%)	0 (0.0%)	12 (11.4%)
**MMP9. Intima**				
0	34 (65.4%)	40 (76.9%)	6 (85.7%)	80 (72.1%)
1	8 (15.4%)	10 (19.2%)	1 (14.3%)	19 (17.1%)
2	7 (13.5%)	2 (3.8%)	0 (0.0%)	9 (8.1%)
3	3 (5.8%)	0 (0.0%)	0 (0.0%)	3 (2.7%)
**MMP9. Media**				
0	32 (61.5%)	13 (25.0%)	7 (100.0%)	52 (46.8%)
1	17 (32.7%)	16 (30.8%)	0 (0.0%)	33 (29.7%)
2	3 (5.8%)	21 (40.4%)	0 (0.0%)	24 (21.6%)
3	0 (0.0%)	2 (3.8%)	0 (0.0%)	2 (1.8%)
**MMP9. Adventitia**				
N-Miss	0	1	5	6
0	26 (50.0%)	6 (11.8%)	2 (100.0%)	34 (32.4%)
1	26 (50.0%)	19 (37.3%)	0 (0.0%)	45 (42.9%)
2	0 (0.0%)	19 (37.3%)	0 (0.0%)	19 (18.1%)
3	0 (0.0%)	7 (13.7%)	0 (0.0%)	7 (6.7%)

**Table 6 biomedicines-12-00619-t006:** Exact numbers of **TIMP-1 and -2** grading data by two independent pathologists for the three groups (mean values from two independent but concordant evaluations).

	A (*n* = 52)	D (*n* = 52)	H (*n* = 7)	Total (*n* = 111)
**TIMP1. Intima**				
N-Miss	0	1	0	1
0	25 (48.1%)	30 (58.8%)	7 (100.0%)	62 (56.4%)
1	12 (23.1%)	14 (27.5%)	0 (0.0%)	26 (23.6%)
2	10 (19.2%)	5 (9.8%)	0 (0.0%)	15 (13.6%)
3	5 (9.6%)	2 (3.9%)	0 (0.0%)	7 (6.4%)
**TIMP1. Media**				
0	26 (50.0%)	9 (17.3%)	7 (100.0%)	42 (37.8%)
1	17 (32.7%)	26 (50.0%)	0 (0.0%)	43 (38.7%)
2	8 (15.4%)	17 (32.7%)	0 (0.0%)	25 (22.5%)
3	1 (1.9%)	0 (0.0%)	0 (0.0%)	1 (0.9%)
**TIMP1. Adventitia**				
N-Miss	0	1	5	6
0	49 (94.2%)	10 (19.6%)	2 (100.0%)	61 (58.1%)
1	3 (5.8%)	18 (35.3%)	0 (0.0%)	21 (20.0%)
2	0 (0.0%)	21 (41.2%)	0 (0.0%)	21 (20.0%)
3	0 (0.0%)	2 (3.9%)	0 (0.0%)	2 (1.9%)
**TIMP2. Intima**				
N-Miss	0	1	0	1
0	39 (75.0%)	51 (100.0%)	7 (100.0%)	97 (88.2%)
1	11 (21.2%)	0 (0.0%)	0 (0.0%)	11 (10.0%)
2	2 (3.8%)	0 (0.0%)	0 (0.0%)	2 (1.8%)
**TIMP2. Media**				
N-Miss	0	1	0	1
0	35 (67.3%)	41 (80.4%)	7 (100.0%)	83 (75.5%)
1	16 (30.8%)	9 (17.6%)	0 (0.0%)	25 (22.7%)
2	1 (1.9%)	1 (2.0%)	0 (0.0%)	2 (1.8%)
**TIMP2. Adventitia**				
N-Miss	0	0	5	5
0	42 (80.8%)	31 (59.6%)	2 (100.0%)	75 (70.8%)
1	10 (19.2%)	15 (28.8%)	0 (0.0%)	25 (23.6%)
2	0 (0.0%)	6 (11.5%)	0 (0.0%)	6 (5.7%)

**Table 7 biomedicines-12-00619-t007:** Evidence of media degeneration on histological sections of the study groups (*p* < 0.271).

	A (*n* = 52)	D (*n* = 52)	H (*n* = 7)	Total (*n* = 111)
N-Miss	0	1	0	1
Yes	26 (50.0%)	31 (60.8%)	0 (0.0%)	57 (51.8%)
No	26 (50.0%)	20 (39.2%)	7 (100.0%)	53 (48.2%)

**Table 8 biomedicines-12-00619-t008:** Evidence of arteriosclerosis on histological sections of the study groups (*p* < 0.100). Classification according to Stary et al., 1995 [4].

	A (*n* = 52)	D (*n* = 52)	H (*n* = 7)	Total (*n* = 111)
0	14 (26.9%)	18 (34.6%)	7 (100.0%)	39 (35.1%)
1	2 (3.8%)	4 (7.7%)	0 (0.0%)	6 (5.4%)
2	14 (26.9%)	16 (30.8%)	0 (0.0%)	30 (27.0%)
3	5 (9.6%)	4 (7.7%)	0 (0.0%)	9 (8.1%)
4	4 (7.7%)	3 (5.8%)	0 (0.0%)	7 (6.3%)
5	9 (17.3%)	6 (11.5%)	0 (0.0%)	15 (13.5%)
6	4 (7.7%)	1 (1.9%)	0 (0.0%)	5 (4.5%)

## Data Availability

All the data and material that support this study are available from the corresponding author upon reasonable request.

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
