# Peer review of "The Role of Matrix Metalloproteinases in Thoracic Aortic Disease: Are They Indicators for the Pathogenesis of Dissections?"

_biomedicines, 2024, doi:10.3390/biomedicines12030619_

Round 1

Reviewer 1 Report

Comments and Suggestions for Authors

This clinical study aims to address three key inquiries: the distribution of MMP/TIMP within the aortic wall's cross-section, the variance in MMP/TIMP patterns between aneurysms and dissections, and insights into the pathophysiology of cardiovascular diseases. However, the articulation of these inquiries was imprecise, and the conclusion lacked clarity. The reviewer has enumerated several points for consideration as follows:

Major comments:

1.       The authors are advised to articulate their objectives with greater clarity in both the abstract and the main text.

2.       The segments from lines 66 to 77, along with Tables 1 to 4, ought to be relocated to the results section, necessitating the elaboration of methodologies in their stead.

3.       In Tables 1 to 3, it is imperative for the authors to specify the parameters compared when presenting "p-values," to address this ambiguity.

4.       The manuscript exhibits inconsistent capitalization of "table" and "figure" throughout. This requires correction to adhere to standard manuscript formatting.

5.       It was observed that Tables 7 and 8 in the supplementary table files were not designated as supplementary. Could the authors clarify if this was intentional?

6.       The conclusion presented lacks clarity. It is essential for the authors to concisely state their conclusions in both the abstract and the main body of the text.

Minor comment:

7.       Portions of the text are italicized. The authors should elucidate the significance of this formatting choice.

8.       The font size in lines 344-350 is noticeably larger than in the rest of the document. This discrepancy should be corrected for uniformity.

Author Response

Review 1

First of all, I would like to thank the reviewer for his acceptance of our review and assure you that we are grateful for any suggestions for improving our manuscript.

According to your comments:

  1. The authors are advised to articulate their objectives with greater clarity in both the abstract and the main text.

We have revised the text of our manuscript and made corrections. We hope for better clarity.

  1. The segments from lines 66 to 77, along with Tables 1 to 4, ought to be relocated to the results section, necessitating the elaboration of methodologies in their stead.

The sentence in lines 66 to 77 was intended as an additional reference and appears to be misleading. We have included this sentence in the results section. The placement of the tables was the publisher's decision and is certainly due to the space available on the pages.

  1. In Tables 1 to 3, it is imperative for the authors to specify the parameters compared when presenting "p-values," to address this ambiguity.

We have corrected. Even the mistakes in table 4 (Thank U!) corrected. Corrected Tables are already inserted in the main text.

Table 1:   Irqsusi et al.

Patients’data (Bold numbers represent significance.)

Aneurysm (n=52)

Dissection (n=52)

Control

(n=7)

Total (n=111)

P value

Gender

male

40 (76.9%)

36 (69.2%)

5 (71.4%)

81 (73.0%)

  0.377

female

12 (23.1%)

16 (30.8%)

2 (28.6%)

30 (27.0%)

  0.377

Age/ Median

65.0 yrs.

60.0 yrs.

78.0 yrs.

64.0 yrs.

  0.011

          Range

40.0 – 84.0 yrs.

28.0 – 82.0 yrs.

51.0 –85.0 yrs.

28.0- 85.0 yrs.

EF

         normal

40 (76.9%)

36 (70.6%)

   4 (80.0%)

80 (74.1%)

  0.337

         reduced

  5 (  9.6%)

  7 (13.8%)

   0 (  0.0%)

12 (11.1%)

  0.539

elective surgery

51 (98.1%)

  1 (1.9%)

  5 (71.4%)

57 (51.4%)

<0.001

emergency

  1  (1.9%)

51 (98.1%)

  2 (28.6%)

54 (48.6%)

<0.001

surgical procedure

valve- type

conduit

10  (19.2%)

  6 (11.5%)

  0 (0.0%)

16 (14.4%)

  0.277

supracoronary

replacement

40 (76.9%)

 31 (59.6%)

  0 (0.0%)

71 (64.0%)

  0.058

supracoronary ascending aorta replacement with arch replacement

  2 (3.8%)

 15 (28.8%)

  0 (0.0%)

17 (15.3%)

<0.001

additional surgical procedures

AVR

21 (40.4%)

   3 ( 5.8%)

  0 (0.0%)

24 (21.6%)

<0.001

AVRepair

13 (25.0%)

 12 (23.1%)

  0 (0.0%)

25 (22.5%)

  0.697

CABG

10 (19.2%)

   7 (13.4%)

  5 (71.4%)

22 (19.8%)

  0.239

Complications

Arrythmia

  9  (17.3%)

   4 (  7.7%)

  0 (0.0%)

13 (11.7%)

  0.138

Bleeding

  7 (13.5%)

 10 (19.2%)

  0 (0.0%)

17 (15.3%

  0.426

Infection

  2 (  3.8%)

   5  (  9.6%)

  0 (0.0%)

  7 (  6.3%)

  0.240

Ischemia

  3 (  5.8%)

   5  (  9.6%)

  0 (0.0%)

  8 (  7.2%)

  0.462

Organ failure

  1 (  1.9%)

   1 (   1.9%)

  0 (0.0%)

  2 (  1.8%)

  1.000

Mortality

  2 (  3.8%)

 11 ( 21.2%)

  0 (0.0%)

15 (13.5%)

  0.008

Table 2:  Irqsusi et al.

Concomitant Diseases in % (Bold numbers represent significance.)

Concomitant Diseases

Aneurysm

Dissection

Control

Total

P Value

Hyperlipidemia

    23.1

      21.2

     28.6

      22.5

  0.813

Arterial Hypertension

    73.1

      75.0

     85.7

      74.8

  0.823

CIHD

    38.5

      19.2

     85.7

      32.4

  0.030

Diabetes mellitus II

    11.5

        3.8

     28.6

        9.0

  0.141

Table 3:  Irqsusi et al.

Echocardiographic measurements of the Aortic roots (data in mm)( Bold numbers represent significance.)

Diameter

Aneurysm

Dissection

Total

P value

Anulus, Median

               Range

        34.6

   22.5 – 50.2

           29.1

     23.2 – 39.9

          30.8

    22.5 – 50.2

    0.006

S.-tubular Junct.,

Median/ Range

        47.4

   32.3 – 61.2

          41.8

     29.4 – 59.0

          45.1

    38.1 – 50.6

    0.272 

Ascending Part

Median/ Range

        50.3

   39.6 – 67.0

          44.4

     33.1 – 59.0

          48.3

    33.1 – 67.0

    0.003

Max. Diameter

Median/ Range

        51.5

   39.6 – 67.0

          46.7

     33.1 – 59.0

          49.6

    33.1 – 67.0

    0.003

  1. The manuscript exhibits inconsistent capitalization of "table" and "figure" throughout. This requires correction to adhere to standard manuscript formatting.

The tables and figures have been labeled with small letters.

  1. It was observed that Tables 7 and 8 in the supplementary table files were not designated as supplementary. Could the authors clarify if this was intentional?

Also with regard to the comments of other reviewers, I would like to suggest, subject to the publisher's acceptance, that all tables and figures are included in the publication on an equal footing. Therefore, the term "supplement" should be omitted.

  1. The conclusion presented lacks clarity. It is essential for the authors to concisely state their conclusions in both the abstract and the main body of the text.

We have revised the text and hope for acceptance.

Minor comment:

  1. Portions of the text are italicized. The authors should elucidate the significance of this formatting choice.

It was made for explanation the terms in the figures or for pronunciation and clarity. Hooping it helps. Not essential!

  1. The font size in lines 344-350 is noticeably larger than in the rest of the document. This discrepancy should be corrected for uniformity.

Corrected.

Reviewer 2 Report

Comments and Suggestions for Authors

The present manuscript compares from the histological point of view, through attributing a specific score to the presence of MMP1, MMP9 and corresponding TiMPs , the relevance of these molecules in the lesioned tissues from patients with abdominal aortic aneurysm and aortic dissection. Moreover the authors use few patient’ samples of morphologically healthy aorta from patients with CAD as controls.

Although interesting the idea of revisiting classic pathological  evaluation in more accurate statistical ways for obtaining novel insights on the pathogenesis of these disorders, there are a number of missing info potentially affecting the results and several issues needing to be improved. 

Comments:

Population:

1.        the control group is deeply unbalanced (7 CAD controls vs. 52 AAA + 52 AD) and I doubt it reach a sufficient statistical power for being compared to the other groups.  

2.        the relative contribution and difference in results between gender is not took in account in the results: if possible perform this analysis, or state the point as a limitation. 

3.        there is no mention about how many (if present) patients with AAA were submitted to surgery because of ruptured aneurysm.

4.        the patient’ entrollment lasted from 2007 to 2015: the. Echocardiographic evaluation methodology and instrumentation was homogenous  andfully superposanble? 

Methods:

In the histology stain description there is no mention of Alcian blue and van Gieson but only of HE.

 The morphometrical analysis it is not described in detail. Basic infos are missing, specifically: 

1.        how many sections are analyzed/marker/sample, 

2.        the sections are cut at what distance (in microns) one from the successive, 

3.        how many positive cells are counted as minimum and/or how many fields are considered/smaple/marker

4.        how are chosen inside the sections the fields to analyze

5.        what is the magnification used in the analysis (in the figures the scale bare is unreadable)

In the settings of aortic atherosclerotic lesions the morphology and composition may change along the aorta lenght and it is compulsory to take them in account for quantitative studies ss the. present one. Please add the information.

The. statement at lines 163-165 should be moved from Results into Methods

Results: 

Histology with Table of disease severity should be moved before IHC  to provide a description of the lesions prior to enter in marker details. It could be useful to show the same areas for both HE and IHC.

In Table 4 and 5 the total number of patient/group is lower (51 vs. 49 and 51 vs. 50, respectively) than that in table 1 (52 vs. 52): why there are missing patients? please rectify or explain the reason of these discrepancies. 

The analysis shown in Table. 5 concerns results by two different pathologist: the scores are presented as mean values from the two independent evaluations? Please specify and precise whether they are concordant.

It could be useful to add a Table associating the atherosclerosis class with marker scores in the different aorta layers for each group.

The CD31 will label not only adventitial microvessels and luminal endothelium but also intramural medial neovascularizations. A correlative co- staining CD31 with MMPs or TIMPs is needed to provide information on the possible relationship between CD31 and the other markers. Please add the data or delete the part about CD31.

Figure 1 shows a technical control. Please add the. corresponding score to each panel and move to Supplementary figures 

Figure 2 please add as supplementary figure a panel with negative controls of the same areas in serial sections. 

 line 316 " It is stimulated by TNFα."  what is referred to?

Discussion:

The discussion is too long: please summarize and focus better. 

Comments on the Quality of English Language

The English is often twisted.  and there are few errors, such as  missing verbs.  It need to be revised.

Author Response

Reviewer 2:

First of all, I would like to thank the reviewer for his acceptance of our review and assure you that we are grateful for any suggestions for improving our manuscript.

The present manuscript compares from the histological point of view, through attributing a specific score to the presence of MMP1, MMP9 and corresponding TiMPs , the relevance of these molecules in the lesioned tissues from patients with abdominal aortic aneurysm and aortic dissection.

I apologize, as already mentioned in the title of the paper, we investigated aortic aneurysms of the ascending aorta.

Moreover, the authors use few patient’ samples of morphologically healthy aorta from patients with CAD as controls.

I apologize again, the term "healthy controls" is intentionally placed in quotation marks in the paper, as patients with CHD can also show morphological changes, whether macroscopically visible or not. This is precisely why this group is important, as arteriosclerotic processes can also be involved in the formation of aneurysms or dissections. Given the superiority of arterial grafts in bypass surgery, which has been proven in numerous studies, a large number of aortic tissue samples is critical.

Although interesting the idea of revisiting  classic  pathological  evaluation in more accurate statistical ways for obtaining novel insights on the pathogenesis of these disorders, there are a number of missing info potentially affecting the results and several issues needing to improved. 

Comments:

Population:

  1. the control group is deeply unbalanced (7 CAD controls vs. 52 AAA + 52 AD) and I doubt it reach a sufficient statistical power for being compared to the other groups. 

First, as I have already mentioned, given the superiority of arterial grafts in bypass surgery, which has been proven in numerous studies, a large number of aortic tissue samples is critical. Secondly, I may refer to Tables 5 and 6. With a minimum number of histological characteristics in the comparison group, sufficient power in the significance test with SPSS is obviously sufficient.

  1. the relative contribution and difference in results between gender is not took in account in the results: if possible perform this analysis, or state the point as a limitation. 

In accordance with your suggestion, we have added a "Limitations" section and pointed this out.

  1. there is no mention about how many (if present) patients with AAA were submitted to surgery because of ruptured aneurysm.

In the original manuscript we have written in the abstract “All patients were surgically operated in the period from 2007 to 2015. “ (line 17 and 18)

  1. the patient’ entrollment lasted from 2007 to 2015: the. Echocardiographic evaluation methodology and instrumentation was homogenous  and fully superposanble? 

Even though our hospital is located in so called "rich Germany", I can assure you that we cannot always acquire the latest medical technology. Determining the aortic orifice area, the diameter of the sinutubular junction, etc. is part of the basic examinations that we carry out both preoperatively and intraoperatively for years. Beside the cardiologist, an experienced anesthesiologist, the surgeon and the assistant look very closely at the morphology of the aortic root, because these results may have a significant influence on the surgical technique chosen (e.g. BENTALL operation, etc.).

Methods:

In the histology stain description there is no mention of Alcian blue and van Gieson but only of HE.

We introduced a small description. 

 The morphometrical analysis it is not described in detail. Basic infos are missing, specifically: 

  1. how many sections are analyzed/marker/sample, 
  2. the sections are cut at what distance (in microns) one from the successive, 
  3. how many positive cells are counted as minimum and/or how many fields are considered/sample/marker
  4. how are chosen inside the sections the fields to analyze
  5. what is the magnification used in the analysis ( in the figures the scale bare is unreadable)

We have introduced the following text: “For morphometrical analysis there was one section per marker/ sample analyzed. Sections for different staining/ immunohistochemistry were obtained as successive slides at a thickness of 2 µm. For an assessment, the whole slide was examined at a low magnification to find the area with highest marker expression. Slides were evaluated for comparison in a 100x magnification in these areas. For a score 1 or higher, we expected to see any staining of at least medium intensity. For a score 2, when evaluating at a field of 100x magnification, we expected a diffuse staining of medium intensity or a focal staining (<50%) of high intensity in the chosen field. A score 3 corresponds to high intensity staining of the marker in at least 50% field diameter.”

In the settings of aortic atherosclerotic lesions the morphology and composition may change along the aorta lenght and it is compulsory to take them in account for quantitative studies ss the. present one. Please add the information.

In fact, the extent of arteriosclerotic lesions along the aorta can change. Likewise, cystic mediadegeneration in the aorta is not ubiquitous. This is certainly a limitation of histologic studies. This circumstance is mentioned under "Limitations". On the other hand, I would like to refer to the clinical facts. The creation of aortocoronary bypasses (the punching cylinders of the connection to the aorta were considered "healthy control") is based on surgical morphological criteria. Of course, hard aortic areas that suggest a plaque structure are not selected. Histological samples are also taken from the resected aortic segment. The extent of resection is determined by macroscopic and technical surgical aspects.

The. statement at lines 163-165 should be moved from Results into Methods.

Done.

Results: 

Histology with Table of disease severity should be moved before IHC  to provide a description of the lesions prior to enter in marker details. It could be useful to show the same areas for both HE and IHC.

Unfortunately, the expert's note leaves many questions unanswered. A histology corresponding to the different stages of aortic insufficiency and aortic stenosis is beyond the scope of this paper. We assume that changes in aortic valve morphology (table 4) cause histological changes in the aortic wall via altered hemodynamics. We have investigated these changes.  Tables 5 and 6 describe a histological marker analysis corresponding to "aneurysm", "dissection" and "healthy controls" independent of the severity of the aortic valve. 

The parallel presentation of HE and IHC certainly has an academic value. The question remains of what value this presentation has for the aim of this work! The focus of the work is on a statistical evaluation of a histological marker analysis and NOT on the specific pathological morphology of the aortic wall. Of course, there is a close relationship, but on the other hand, a framework must also be drawn. Examples of arteriosclerotic changes and medial degeneration that we have found have been shown (fig. 4 and 5).  The occurrence of arteriosclerosis (Stary- classification) and medial degeneration does NOT differ between the groups (table 7 and 8).

In Table 4 and 5 the total number of patient/group is lower (51 vs. 49 and 51 vs. 50, respectively) than that in table 1 (52 vs. 52): why there are missing patients? please rectify or explain the reason of these discrepancies. 

Please see remark table 4: *in the aortic aneurysmatic group 1 (1.9%) and in case of dissection 2 (3.8%) of the patients had already a preexsisting aortic valve replacement. In the „healthy“ group no insufficiency or stenosis of the tricuspid valves were found.

The analysis shown in Table. 5 concerns results by two different pathologist: the scores are presented as mean values from the two independent evaluations? Please specify and precise whether they are concordant.

Of course, these are two independent evaluations whose values matched. We have added in the text.

It could be useful to add a Table associating the atherosclerosis class with marker scores in the different aorta layers for each group.

Of course, a further correlation study between the manifestation of arteriosclerotic changes in the aortic wall and MMP and TIMP expression is desirable. However, since this expression is to be understood as a reaction in response to mechanical stress, this examination should definitely be correlated with a hemodynamic-functional study (echo, CT scan for different strains of the aortic wall). We have already tackled this problem (see below) and will continue to do so. However, it is undoubtedly beyond the scope of this paper.

Vogt S, Karatolios K, Wittek A, Blasé C, Ramaswamy A, Mirow N, Moosdorf R. Detailed Measurement of Wall Strain with 3D Speckle Tracking in the Aortic Root: A Case of Bionic Support for Clinical Decision Making. Thorac Cardiovasc Surg Rep. 2016 Dec;5(1):77-80. doi: 10.1055/s-0036-1571815. 

Karatolios K, Wittek A, Nwe TH, Bihari P, Shelke A, Josef D, Schmitz-Rixen T, Geks J, Maisch B, Blase C, Moosdorf R, Vogt S. Method for aortic wall strain measurement with three-dimensional ultrasound speckle tracking and fitted finite element analysis. Ann Thorac Surg. 2013 Nov;96(5):1664-71. doi: 10.1016/j.athoracsur.2013.06.037. 

The CD31 will label not only adventitial microvessels and luminal endothelium but also intramural medial neovascularizations. A correlative co- staining CD31 with MMPs or TIMPs is needed to provide information on the possible relationship between CD31 and the other markers. Please add the data or delete the part about CD31.

Even if reviewer 2 is an obviously expert in the assessment of tissues (pathologist?), I have to thoroughly contradict my colleague on this point. CD 31 reacts with a 130 kDa glycoprotein, also designated platelet endothelial cell adhesion molecule-1 (PECAM-1). The antibody strongly labels endothelial cells. So,  when  CD31 will label not only adventitial microvessels and luminal endothelium but also intramural medial neovascularizations it correlates with our intension to show vascularization at all. We say, “in atherosclerosis, neovascularization occurs mainly in the intima, whereas in medial degeneration and case of dissection, many vessels can be detected either in the media or adventitia, but not in the intima.” The target was an “all or nothing”. It was not our intention to provide evidence of gradual neorevascularization or its repression. However, it was impressive for us to observe that a "yes/no" vascularization occurs histologically in the case of aneurysm/dissection at the intima/media sites. From our point of view, this is highly interesting and should therefore also be communicated. If the CD 31 antibody detects not only vascularization but also neovascularization, this antibody appears to be a particularly powerful tool. For this very reason, we have explained in the discussion:

“TIMP family members have the ability to suppress the proliferation of endothelial cells, directly. This protein may be critical to the maintenance of tissue homeostasis by suppressing the proliferation of quiescent tissues in response to angiogenic factors, and by inhibiting protease activity in tissues undergoing remodelling of the extracellular matrix [19, 20, 21]. Interestingly, the highest TIMP2 expression was measured in the intima in case of aneurysms. TIMP2 was less expressed in the case of dissection.”

So, we SUGGESTED: “It can be stated that in the investigated aortic tissue a clear proteolytic transformation can be assumed, which favors a progressive destruction of the extracellular matrix, but also degradative and constructive (reparative) processes can be assumed, focussed on angiogenesis and neoangiogenesis of the supplying blood vessels (vasa vasorum) [2, 22].”

Further work is planned and cooperation is welcome. An imperative “Please add the data or delete the part about CD31.” is from our point of view not justified.

Figure 1 shows a technical control. Please add the corresponding score to each panel and move to Supplementary figures

Unfortunately, I must also confess my restricted understanding of your question at this point. In Figure 1 we have shown control sections of 4 tissues to demonstrate the clear reactivity (yes/no) of the relevant antibodies. For these control sections, tissues were used that are known to express the corresponding markers even under physiological conditions. However, the quantity of expression is dependent on the tissue! Thus, a score system can only be specific to tissue and would therefore be irrelevant for the aorta. The fact that the different antibodies have gradually different reactions is shown per se by the different scores in Figure 2, which means that the requirement for a score in Figure 1 is irrelevant. We have also included all figures and tables in the publication with regard to the statements of other experts.

Figure 2 please add as supplementary figure a panel with negative controls of the same areas in serial sections. 

At this point, I would like to reply with all due respect and point out that it is difficult for me to answer this question, which seems difficult to understand.

In figure 1, extra tissues were selected that are known to permanently express the corresponding markers. Thus, we have positive controls for the affinity of the selected antibodies. The antibodies were applied under the same, almost identical conditions. The different expression of the investigated markers in the aortic tissue is proven by the fact that with different pathological findings of the aorta, both almost no (score 0) and massive expression (score 3) occurs.  Therefore, if someone wanted to present an exactly "clean" and truly negative sample in "serial sections" for each score and each marker, there is only one possibility left: while maintaining the corresponding age interval (see Table 1), someone would have to persuade completely healthy subjects to undergo heart surgery in order to take corresponding tissue samples. You can always find degenerative changes of varying degrees in this age interval. An intraoperative "search" for healthy aortic vessels in case of a CIHD, aneurysm or an aortic dissection (emergency case!) appears “questionable.”

line 316 " It is stimulated by TNFα."  what is referred to?

The sentence in line 316 belongs in the discussion section and has been deleted.

Discussion:.

The discussion is too long: please summarize and focus better. 

We have revised the text and hope for acceptance.

Round 2

Reviewer 1 Report

Comments and Suggestions for Authors

This clinical study was revised but it was not enough in current form. This reviewer has several comments as follows:

Major comments:

1.       The objective (hypothesis) has been included in the abstract but is absent from the main text. The authors should incorporate the hypothesis towards the conclusion of the Introduction section.  

2.       Regarding Tables 1-4, while the authors have conducted statistical analyses and presented p-values, it is recommended that they explicitly specify the parameters that were compared in the footnotes - presumably among aneurysm, dissection, and control groups, excluding the total. Furthermore, it is advisable to relocate these tables (1-4) to the Results section for clarity and coherence. 

Author Response

I thank You for your comment and help.

We comment the reviewer’s remarks as follows:

  1. The objective (hypothesis) has been included in the abstract but is absent from the main text. The authors should incorporate the hypothesis towards the conclusion of the Introduction section.  

We thank you for the important hint and have inserted the following sentence in the introductory text: The investigations of metalloproteases and their inhibitors should provide information about the remodeling process of the ECM. It is assumed that this is not uniform and influences neoangiogenesis in different ways. It may result in disturbed tissue nutrition, which appears to be the basis for the development of aortic dissection.

  1. Regarding Tables 1-4, while the authors have conducted statistical analyses and presented p-values, it is recommended that they explicitly specify the parameters that were compared in the footnotes - presumably among aneurysm, dissection, and control groups, excluding the total.

We have marked the significant values with asterisks in Tables 1 - 4.

Furthermore, it is advisable to relocate these tables (1-4) to the Results section for clarity and coherence. 

We would like to point out that we have in fact listed tables 1 to 4 under the heading "Results" (see lines 151 to 162). The article has certainly been typeset by the editor for reasons of organization, but this does not mean that the tables have been moved to the "Material and methods" section.

Reviewer 2 Report

Comments and Suggestions for Authors

dear Authors,

I continue to have different opinion from yours about several not critical points, but I thinks that it is a problem of point of views and scientific approach. The manuscript can be considered as acceptable in the revisioned version.

thank you for the comments

Author Response

I thank You for the comments.

For Reviewer 2:

I think no corrections needed.

Round 3

Reviewer 1 Report

Comments and Suggestions for Authors

The authors do not understand this reviewer’s comments.